# Reassessing the adrenomedullin scavenging function of ACKR3 in lymphatic endothelial cells

Elena C. Sigmund[1], Aline Bauer[1], Barbara D. Jakobs[2], Hazal Tatliadim[1], Carlotta Tacconi◯[1], Marcus Thelen[3], Daniel F. Legler[2,4], Cornelia Halin◯[1]*

1 Institute of Pharmaceutical Sciences, ETH Zurich, Zurich, Switzerland, 2 Biotechnology Institute Thurgau (BITg), University of Konstanz, Kreuzlingen, Switzerland, 3 Institute for Research in Biomedicine (IRB), Università della Svizzera Italiana, Bellinzona, Switzerland, 4 Theodor Kocher Institute, University of Bern, Bern, Switzerland

* cornelia.halin@pharma.ethz.ch

**Data Availability Statement:** All relevant data are within the paper and its Supporting Information files. Raw data are available on the ETH research collection (http://hdl.handle.net/20.500.11850/

## Abstract

Atypical chemokine receptor 3 (ACKR3) is a scavenger of the chemokines CXCL11 and CXCL12 and of several opioid peptides. Additional evidence indicates that ACKR3 binds two other non-chemokine ligands, namely the peptide hormone adrenomedullin (AM) and derivatives of the proadrenomedullin N-terminal 20 peptide (PAMP). AM exhibits multiple functions in the cardiovascular system and is essential for embryonic lymphangiogenesis in mice. Interestingly, AM-overexpressing and ACKR3-deficient mouse embryos both display lymphatic hyperplasia. Moreover, *in vitro* evidence suggested that lymphatic endothelial cells (LECs), which express ACKR3, scavenge AM and thereby reduce AM-induced lymphangiogenic responses. Together, these observations have led to the conclusion that ACKR3-mediated AM scavenging by LECs serves to prevent overshooting AM-induced lymphangiogenesis and lymphatic hyperplasia. Here, we further investigated AM scavenging by ACKR3 in HEK293 cells and in human primary dermal LECs obtained from three different sources *in vitro*. LECs efficiently bound and scavenged fluorescent CXCL12 or a CXCL11/12 chimeric chemokine in an ACKR3-dependent manner. Conversely, addition of AM induced LEC proliferation but AM internalization was found to be independent of ACKR3. Similarly, ectopic expression of ACKR3 in HEK293 cells did not result in AM internalization, but the latter was avidly induced upon co-transfecting HEK293 cells with the canonical AM receptors, consisting of calcitonin receptor-like receptor (CALCRL) and receptor activity-modifying protein (RAMP)2 or RAMP3. Together, these findings indicate that ACKR3-dependent scavenging of AM by human LECs does not occur at ligand concentrations sufficient to trigger AM-induced responses mediated by canonical AM receptors.

610801). The DOI is as follows: 10.3929/ethz-b-000610801

**Funding:** Swiss National Science Foundation (https://www.snf.ch/en) Sinergia program (CRSII3_160719 / 1) for CH, MT, DF (Cornelia Halin, Marcus Thelen, Daniel Legler) ETH Zurich Core Funding for CH (Cornelia Halin)

**Competing interests:** The authors have declared that no competing interests exist.

# Introduction

ACKR3, formerly known as CXCR7 [1] or RDC1 [2], is an atypical chemokine receptor expressed by blood vascular and lymphatic endothelial cells (BECs and LECs, respectively) amongst many other cell types [3, 4] and is a well-known scavenging receptor of the chemokines CXCL12 and CXCL11 [5]. Rather than inducing G protein-mediated signaling, ligand binding to ACKR3 triggers β-arrestin recruitment and internalization of the receptor/ligand complex, followed by intracellular ligand degradation [3, 4]. By regulating site-specific or systemic concentrations of CXCL12, ACKR3 was shown to impact diverse biologic processes, including the migration of leukocytes [6] and tumor cells [7] as well as of germ cells during development [8]. Recent studies additionally revealed that ACKR3 functions as a broad-range scavenging receptor for a class of opioid peptides [9, 10]. In mice, ACKR3-deficiency was found to be embryonically lethal as it results in abnormal cardiac development and hyperplasia, in addition to abnormalities in neuronal development and in the immune system [11–13]. Moreover, ACKR3-deficient embryos display lymphatic hyperplasia and lymphedema [14], a phenotype that has been attributed to ACKR3's role as a scavenging receptor for AM in LECs [14]. AM is a well described vascular peptide hormone and driver of endothelial cell proliferation [14]. AM was shown to induce proliferation, migration and tube formation of LECs *in vitro* [15], as well as lymphangiogenesis and modulation of lymphatic drainage in mice *in vivo* [15, 16]. AM signals via its conventional receptors AM1 and AM2, which are heterodimers consisting of the calcitonin receptor-like receptor (CALCRL) in association with either the receptor activity-modifying protein (RAMP)2 (AM1) or RAMP3 (AM2) [17].

Interestingly, upon crossing ACKR3$^{-/-}$ animals with AM haplo-insufficient (Adm$^{+/-}$) animals, Klein et al. observed that AM haploinsufficiency improved postnatal survival and rescued the phenotype caused by ACKR deficiency in ACKR3$^{-/-}$ Adm$^{+/-}$ animals. In contrast, postnatal survival was significantly reduced in ACKR3$^{+/-}$ mice crossed with AM overexpressing Adm$^{hi/hi}$ mice [14]. The observed phenotypic interconnection between ACKR3 and AM expression *in vivo* was in agreement with a previous literature report [2] suggesting that ACKR3 functions as a scavenging receptor for AM. In further direct support of this assumption, knockdown of ACKR3 in LECs *in vitro* was shown to enhance AM-induced proliferation and migration of human LECs [14]. Moreover, in indirect support of ACKR3-mediated AM scavenging, ectopic expression of ACKR3 in HEK293T cells was found to enhance the depletion of biotinylated AM from the cell culture supernatant, whereas this process was diminished upon shRNA-mediated knockdown of ACKR3 in human LECs [14]. Based on these findings [14] it is nowadays commonly assumed in the field of lymphatic biology that ACKR3 expression in LECs serves to counterbalance overshooting AM-induced lymphangiogenic responses [18–22]. However, studies performed in ACKR3 overexpressing human cell lines have recently questioned the ability of AM to be scavenged by ACKR3 at physiologically meaningful concentrations [23, 24]. For example, Meyrath et al. could only detect β-arrestin recruitment to ACKR3 in response to sub-micromolar AM concentrations in HEK293 cells transfected with ACKR3 [24]. Moreover, a previous study from our lab found that conditional, postnatal knockout of ACKR3 in the lymphatic vasculature in mice did neither recapitulate the observations made during embryonic development nor affect lymphatic function [25].

In the present study, we therefore set out to better investigate the ability of LEC-expressed ACKR3 to scavenge AM and to modulate LEC responses to AM. To this end, we performed *in vitro* LEC proliferation assays as well as uptake assays with fluorescently labeled AM and chemokine in primary human dermal neonatal, juvenile or adult LECs. Notably, three different sources of LECs were used to account for potential differences in ACKR3 expression and scavenging activity between LEC donors and/ or differences caused by the donors' age. ACKR3

function was either pharmacologically modulated using the ACKR3-specific competitive agonist CCX771, a well-described inhibitor of CXCL12 scavenging [7, 26], or knocked-down by shRNA. In addition, we studied AM scavenging in HEK293 cells transfected with ACKR3 and/or different components of the canonical AM receptors. These experiments did not provide any evidence of ACKR3 mediating the scavenging of AM.

## Material and methods

### Isolation and culture of adult primary human skin LECs (adLECs)

LECs were isolated and cultured as previously described [27]. Briefly, LECs were obtained from the breast skin of a healthy 58 years old female subject admitted for plastic surgery at the University Hospital Zurich. Written informed consent was obtained as approved by the Ethics Committee of the Kanton Zurich (2017–00687). The skin sample was washed in hank's balanced salt solution (HBSS) supplemented with 5% fetal bovine serum (FBS, Gibco), 2% antibiotic and antimycotic solution (AA, Gibco), and 20mM HEPES (Gibco), and incubated in 0.25% trypsin (Sigma) overnight at 4˚C. After removal of the epidermal sheets, the dermis was finely minced and enzymatically digested (RPMI medium, 5% FBS, 2% AA, 20mM HEPES, 1000U/mL collagenase type 1 (Worthington), 40μg/mL DNase I (Roche) for 45 min at 37˚C under constant agitation. The digested tissue was then filtered through a 100μm cell strainer (Falcon), washed with RPMI medium supplemented with 10% FBS, 2% AA, and 20mM HEPES, and centrifuged at 485 ×g for 6min at 4˚C. Cells were seeded into plates previously coated with 10 μg/ml fibronectin (Roche) and cultured in EGM-2 complete medium (Lonza) supplemented with 5% FBS at 37˚C in a 5% $CO_2$ incubator. After 7–10 days, cells were trypsinized, and endothelial cells were selected based on CD31 positivity with Dynabeads CD31 magnetic beads (Thermo Fisher Scientific) and cultured until confluency. Endothelial cells were detached, washed with FACS buffer (DPBS supplemented with 2% FBS and 1mM EDTA), and incubated with Alexa647-conjugated mouse anti-human podoplanin antibody (1:70, clone 18H5, Novus Biologicals) and PE-conjugated mouse anti-human CD31 antibody (1:20, clone WM59, BD Pharmingen) in FACS buffer for 30min at 4˚C. After a wash with FACS buffer, adLECs were finally sorted according to their positivity for CD31 and podoplanin on a FACSAria II (BD Biosciences) with a 100μm nozzle, using FACSDiva software (ver. 6.1.3). adLECs were used between passage (p) 2 and 6. Cells were routinely tested for mycoplasma contamination using the MycoScope PCR Mycoplasma Detection Kit (Genlantis).

### Culture of commercial primary human neonatal and juvenile dermal LECs (ndLECs and jdLECs)

Single donor human neonatal dermal lymphatic microvascular endothelial cells (ndLECs) were obtained from Lonza (Visp, Switzerland, catalogue #CC-2812; HMVEC-dLyNeo) and cultured in complete EBM-2 medium consisting of EBM-2 basal Medium (CC-3156, Lonza) supplemented with EGM-2 MV Single Quots™ (CC-4147, Lonza). Cell culture dishes were pre-coated with 10μg/ml collagen type 1 (Advanced Biomatrix) and 10μg/ml fibronectin (Merck Millipore) in PBS.

Juvenile skin LECs (jdLECs) from the foreskin of a single male donor (Lot Nr. 431Z006.2) were obtained from Promocell (Heidelberg, Germany, # C-12216) and cultured in complete EBM-2 (CC-3156, Lonza) supplemented with EGM™-2 SingleQuots™ (CC-4176, Lonza), without VEGF-A (CC-4176, Lonza), additionally supplemented with (3%) FBS for a final concentration of 5% FBS (Gibco), as used for ndLECs. Cell culture dishes were pre-coated with

10μg/ml collagen type 1 (Advanced Biomatrix, San Diego, CA, USA) and 10μg/ml fibronectin (Merck Millipore) in PBS.

## HEK293 cell culture

HEK293-CTRL and stably transfected HEK293-ACKR3 cells (generated as described below) were maintained in DMEM (Invitrogen), 10% FBS (Gibco) 1% penicillin/streptomycin (Gibco).

## qRT-PCR analysis of *ackr3*, *ramp2*, *ramp3* and *calcrl* expression in LECs and HEK293 cells

Total cellular RNA was extracted from confluent plates of human LECs or HEK293 control (HEK293-CTRL) cells and HEK293 cell stably transfected with ACKR3 (HEK293-ACKR3) using TRIZOL reagent (Life Technologies, cat. no.15596026). cDNA was generated using the High-Capacity cDNA Reverse Transcription Kit (Fisher Scientific, cat. no. 10400745). Quantitative PCR was performed on reversely transcribed RNA using PowerUp SYBR Green Master Mix (Thermo Fisher Scientific, cat. no. A25776). Gene expression analysis was performed on a QuantStudio 7 Flex system (Applied Biosciences) using the following primer pairs:

*Ackr3* (RV: 5'-GTA GAG CAG GAC GCT TTT GTT-3'; FW: 5'-TCT GCA TCT CTT CGA CTA CTC- 3'); *Calcrl* (RV: 5'-CAT CAA TGG TGT GCT GGA AC-3'; FW: 5'-CAC TAT GCC TGA TGT GAC GC-3'); *Ramp2* (RV: 5'-GTT GGC AAA GTG GAT CTG GT- 3'; FW: 5'-GCC ATG ATT AGC AGG CCT TA-3'); *Ramp3* (RV: 5'-CTC ATC CCG CTG ATC GTT AT- 3'; FW: 5'AAC TTT CTT CCA GCT TGC CA- 3'); *β-actin* (ACTB) (QuantiTect primer, Qiagen, cat no. QT01680476). *Cxcr4* (QuantiTect primer, Qiagen, cat no. QT00223188).

## Anti-ACKR3 staining of cultured human LECs for flow cytometry

*In vitro* cultured ndLECs were washed 2x with PBS and detached with Accutase (A6964, Sigma-Aldrich) at 37°C. Antibody staining with 5μg/ml monoclonal mouse anti-human ACKR3-APC (clone 11G8, originally described as anti-CXCR7/RDC-1-APC, R&D) or 10μg/ml mouse anti-human ACKR3 (clone 9C4, originally described as anti-CXCR7 [28]) or corresponding isotype controls (mouse IgG1-APC and mouse IgG1) was performed in FACS buffer for 30min on ice. Mouse anti-human CXCR7 (9C4) was detected with a secondary anti-mouse IgG Alexa Fluor488 antibody (Invitrogen) and acquired on a *FACSCanto™* (BD Biosciences). The antibody staining with anti-human ACKR3-APC (clone 11G8) was acquired on a Cytoflex S apparatus.

## ShRNA-mediated silencing of ACKR3 in human LECs

Briefly, adLECs p2 or jdLECs p3 were seeded and expanded for 4 days in a 10 cm culture plate in EBM2 complete medium (Lonza). On the day of the transduction with lentivirus particles, containing one of four different shRNAs or the scrambled control, LECs cells were seeded into 5x 4 wells of a 6-well plate at a concentration of 50 000 cells/ well. (Four wells for each of the ACKR3 shRNAs (A-D) and the scramble control, packaged into Lentivirus particles). After 5h, LECs were infected with ACKR3- shRNA Lentivirus particles (A-D) using a calculated MOI of 10. Four days after transduction, cells were harvested and seeded into a T75 cell culture flask for expansion. Eight days after the Lentivirus infection, LECs were harvested and GFP positive cells were sorted on a FACSAria II (BD Biosciences) cell sorter with a 100μm nozzle, using FACSDiva software (version 6.1.3). Afterwards, cells were expanded and cryopreserved at p6, as a stock for subsequent experiments. Experiments were performed at p7 and p8. The

following commercially available shRNA constructs in pGFP-C-shLenti-particles (Origene) were used for shRNA mediated silencing of ACKR3:

TL305345-**A** (*TI321373)*: *CCGAGCACAGCATCAAGGAGTGGCTGATC*

*TL305345-**B** (TI321374)*: *GAGTGGCTGATCGGCATGGAGCTGGTCTC*

*TL305345-**C** (TI321375)*: *GACACGCACTGCTACATCTTGAACCTGGC*

*TL305345-**D** (TI321376)*: *CGCAACTACAGGTACGAGCTGATGAAGGC*

## Generation of HEK293-ACKR3 cells

HEK293 cells were stably transfected with a hemagglutinin (HA)-tagged ACKR3 construct [28] cloned into pcDNA3. Transfection was performed using lipofectamine according to the manufacturer's instructions (Invitrogen, 153380100). Bulk ACKR3-expressing cells were selected with G418 (0.9 μg/ml, Invitrogen) and tested for HA surface expression, using anti-HA antibody clone 12CA5, and stably transfected HEK293-ACKR3 cells isolated by FACS sorting on a FACSAria™ III (BD Biosciences).

## Plasmid construction for transient expression of CALCRL, RAMP2 and RAMP3 in HEK293 cells

The entire open reading frame of CALCRL was amplified by PCR using the primers 5′–CGCGGATCCGCCATGGAGAAAAAGTGTACCC and 5′–CGCGGATCCCTAATTATATAAAT TTTCTGGTTTTAAGAG and cloned into the EcoRI and ApaI restriction sites of pEGFP-N1 (Clontech), revealing pCALCRL-EGFP. The vector pcDNA3.1(+)_CD33_FLAG-RAMP1 containing a CD33 signal peptide and a FLAG-tag in front of the mature protein sequence of RAMP1 [29] was kindly provided by Dr. David R. Poyner (Aston University, Birmingham, UK) and served as template. The EcoRI restriction site in front of the CD33 signal sequence within the multiple-cloning-site was removed by site-directed mutagenesis. Subsequently, the mature protein sequence of RAMP1 was replaced by the mature protein sequence of RAMP2 and RAMP3 using specific PCR products and the two remaining EcoRI restriction sites. The following two PCR primer pairs were used: for *Ramp2*: 5′– CGATTAGAATTCTTGCATGG ATCCGCTCAGCCTCTTCCC and 5′– CGATTAGAATTCCTAGGCCTGGGCCTCACTG; for *Ramp3*: 5′–GCATTAGAATTCTTGCATGGATCCGCAGGCGGCTGCAACG and 5'–CGATTA GAATT′TTCACAGCAGCGTGTCGGTG. The resulting plasmids pcDNA3.1(+) CD33_FLA-G-RAMP2 and pcDNA3.1(+)_CD33_FLAG-RAMP3 encode for FLAG-tagged RAMP2 and FLAG-tagged RAMP3, respectively.

## Transient transfection of HEK293 cells

For transient transfection of RAMP2, RAMP3 and CALCRL, HEK293-CTRL or HEK293-ACKR3 cells were seeded into a 12 well plate at 125.000 cells / well in 0.6 mL medium and transfected the next day, after reaching 60–80% confluency. Briefly, after addition of fresh medium to each well, the transfection mixture containing 500ng plasmid DNA and 1.875μl *Trans*IT®-LT1 Transfection Reagent (Mirus Bio LLC, Madison USA) in 50μl Opt-MEM™ (Gibco), was added in a drop-wise manner to the cells, according to the manufacturer's instructions. Experiments were performed on day 2 after transfection, when peak induction of CALCRL-eGFP was observed by fluorescent microscopy. Following plasmids were used for

transfection: pcDNA3.1+_CD33_FLAG-RAMP2 cloned according to [30], pCALCR-L-EGFP-N1, pcDNA3.1+_FLAG-RAMP3.

## Use of LECs for different types of experiments

**ndLECs** were used for chemokine uptake assays, anti-ACKR3 staining by flow cytometry, as well as fluorescent AM, CXCL12 or CXCL11/12 uptake experiments analyzed by flow cytometry and microscopy. ndLECs were also used in proliferation assays in combination with CCX771 treatment and AM titration.

**adLECs and jdLECs** were used for all shRNA-mediated ACKR3-knockdown and related assays, including fluorescent AM or CXCL11/12 uptake experiments analyzed by microscopy or flow cytometry and proliferation assays.

## Fluorescently labelled AM and chemokines used in uptake experiments

AFdye568 (AF568)-labelled AM (AM-AF568) was custom-synthesized and fluorescently labelled at Cambridge Research Biochemicals Ltd (Billingham, UK), according to a previous report by Schönauer et al. [31]. In the latter, carboxytetramethylrhodamine (Tam)-labelled AM was successfully used to visualize AM internalization in transfected HEK293 cells [31]. Here, Tam was replaced by AF568, due to experimental considerations and superior dye properties. The amino acid sequence (1–52) of the synthesized AM peptide was: YRQSMNNF-Pra-GLRSFGCRFGTC TVQKLAHQIYQFTDKDKDNVAPRSKISPQGY-amide. Notably, Pra at position 9, which corresponds to the unnatural amino acid L-propargyl glycine, was inserted and replaced a Gln (Q), to allow for the site-specific attachment of AF568 picolyl azide via a copper(I)-catalyzed azide alkyne cycloaddition at this position. In analogy to natural human AM, the resulting AF568-labelled AM peptide [Pra$^9$(AF568)] AM(1–52) contained one intramolecular disulfide bridge between residues $C^{16}$ and $C^{21}$ and an amidated tyrosine at the carboxy terminus.

AlexaFluor 647-labeled CXCL12 (CXCL12-AF647) was purchased from ALMAC Group (UK). The ACKR3-specific chimeric chemokine CXCL11/12 site-specifically labeled with either AlexaFluor 647 (CXCL11/12-AF647) or Atto565 dye (CXCL11/12-Atto565), as previously described in [32].

## Chemokine and AM uptake assays in LECs

Human LECs were seeded into 24-well plates coated with 10μg/ml collagen type 1 (Advanced Biomatrix) and fibronectin (Merck Millipore) (in PBS) in their respective EBM-2 complete culture medium (without human VEGF-A), as described before. After 24h, the medium was changed to starvation medium consisting of EBM-2 basal medium (CC-3156, Lonza), containing 2% FBS (Gibco) and 1% antibiotic-antimycotic solution (Thermo Fisher Scientific) for 24h. For cytokine stimulation (stim), starved LECs were subsequently stimulated for 24h with 20ng/ml TNFα (AF-300-01A, Peprotech, London, UK) and IFN-γ (AF-300-02, Peprotech). When treated with CCX771, LECs were pre-incubated for 30min with 1μM CCX771 (ChemoCentryx, Mountain View, CA, USA) or vehicle control, both diluted in starvation medium, before adding fluorescently-labelled chemokines or AM. For all assays, fluorescent chemokines (50nM) or AM (concentrations as indicated in the text) were diluted in EBM-2 starvation medium and incubated with LECs at 37˚C, in presence of either 1μM CCX771 or the vehicle control. Chemokine/AM binding controls were incubated for 1h at 4˚C. After 1h, LECs were washed once with PBS and subsequently incubated for 1min with an acidic wash buffer (100mM NaCl, 50 mM glycine in PBS, adjusted to pH3) at RT. Afterwards, LECs were washed with PBS and detached from the plates by short incubation with Accutase® at 37˚C. LECs were harvested in ice cold FACS buffer for subsequent acquisition on a Cytoflex S apparatus.

## Ligand competition assay in LECs

Ligand competition assays were performed similar to scavenging assays (see chemokine uptake assays). Indicated molar concentrations of AM (0-10000nM) and 50nM CXCL11/12-AF647 were added simultaneously to cultured ndLECs. After 1h, ndLECs were washed with PBS and subsequently washed shortly with an acid wash buffer (100mM NaCl, 50mM glycine in PBS, adjusted to pH3) and detached using Accutase. Subsequently, ndLECs were harvested in FACS buffer and acquired on a Cytoflex S apparatus.

## LEC proliferation assays

ShRNA-treated adLECs and jdLECs, which were used in case of proliferation assessment upon CCX771 treatment, were seeded into collagen and fibronectin-coated 96-well plates (50μg/ml in PBS) (2200 cells per well, 10 wells per condition). After 24h, the culture medium was changed to starvation medium (EBM-2 and 2% FBS, 1% P/S) for 24h before treatment with 1nM AM (or 0.01nM AM, 0.1nM AM or 10nM AM) (Bachem AG, Switzerland) in starvation medium and optionally in presence of either 1μM of CCX771 or vehicle control (ChemoCentryx, Mountain View, CA, USA), respectively. After 72h, cells were washed with PBS and incubated with 1mg/ml 5-methylumbelliferylheptanoate (MUH, Sigma-Aldrich) for 1h. The fluorescent intensity correlating with the number of viable cells was quantified spectrophotometrically using a SpectraMax Gemini EM microplate reader (Bucher Biotec AG, Basel, Switzerland).

## AM-AF568 binding/internalization experiments in HEK293 cells

AM-AF568 binding/internalization experiments were performed 48h post transfection. Briefly, transfected HEK293-CTRL and HEK293-ACKR3 cells were incubated for 1h in starvation medium (DMEM 2%FBS, 1% P/S) containing 50nM AM-AF568 at either 37˚C or 4˚C. Subsequently, the cells were washed once with PBS and then incubated for 1 min with an acidic wash buffer (100mM NaCl, 50 mM glycine in PBS, adjusted to pH3) at RT. Immediately after, the cells were washed once with PBS, detached with Accutase® at 37˚C, and then harvested in ice cold FACS buffer. One part was used for direct acquisition on a Cytoflex S apparatus while the remaining cells were used for subsequent staining with monoclonal anti-FLAG BioM2 (CCF9291, Sigma-Aldrich), diluted 1:400, and Streptavidin-Brilliant Violet 421™ (CC 405226, Biolegend) diluted 1:300, before acquisition.

## Statistical data analysis

Statistical analysis was performed using Prism 8 (GraphPad Software, LaJolla, CA, USA). All data are presented as mean ± standard deviation (SD). Student's *t*-test (paired, two-tailed) was used to compare the means of two groups. One-way or two-way ANOVA, was used to compare the means of two or more groups and different conditions, followed by either Tukey's post-hoc test, to make individual pairwise comparisons between all groups (and conditions), or alternatively, Šídák's post hoc test, when pairwise comparisons to respective (untreated) controls were calculated. Data that contain mean or matched data from repeated experiments were analyzed using repeated-measure (RM) one-way or two-way ANOVA, followed by either Tukey's post-hoc test, Dunnett's (when comparing all mean values to one shared control value), or Šídák's multiple comparison test. In case of missing repeated data points, a mixed effects model analysis followed Tukey's or Šídák's multiple comparison post-hoc test was performed, A p-value of p<0.05 was considered significant.

## Results

### AM fails to reduce the chemokine scavenging activities of ACKR3 in a ligand competition assay in LECs *in vitro*

For our experiments we used primary dermal LECs isolated from three different sources: namely, LECs derived from neonatal, juvenile or adult human dermis (ndLECs, jdLECs and adLECs, respectively). These cells uniformly expressed CD31 and podoplanin in FACS (**S1 Fig**). qRT-PCR analysis indicated expression of *Ackr3* in steady state and its induction upon stimulating LECs by overnight incubation in presence of the inflammatory cytokines TNFα and IFN γ (**S1 Fig**). Flow cytometry analysis using two different anti-human ACKR3 antibody clones (9C4 [28]) and 11G8 [33]) revealed only minute or non-detectable ACKR3 surface expression in ndLECs under regular culture conditions (**S1 Fig**), likely reflecting the fact that ACKR3 continuously internalizes and recycles through intracellular vesicles [34]. Conversely, in line with the qRT-PCR data, robust ACKR3 surface expression was observed upon over-night stimulation of ndLECs with TNFα and IFNγ (**S1 Fig**).

   To assess the scavenging capacity of LEC-expressed ACKR3, we incubated TNFα/IFNγ stimulated ndLECs for 1h at either 37˚C with fluorescently labelled CXCL12 (CXCL12-AF647), performed an acidic wash to remove surface-bound chemokine and subsequently analyzed the cells by flow cytometry. As a control, chemokine uptake was simultaneously performed at 4˚C, i.e. a temperature at which endocytic processes are strongly impaired. Flow cytometric analysis revealed a high Mean Fluorescent Intensity (MFI) of CXCL12-AF647 upon incubation at 37˚C, indicative of cellular uptake, as compared to a low MFI observed upon incubation at 4˚C (**Fig 1A and 1B***)*. By contrast, when the experiment at 37˚C was performed in presence of the ACKR3-specific competitive agonist CCX771, a well-described inhibitor of CXCL12 scavenging [7, 26], CXCL12-AF647 uptake was markedly reduced (**Fig 1A and 1B**). Similar scavenging experiments were also performed with an Atto565-labelled version of the recombinant chimeric chemokine CXCL11/12 (CXCL11/12-Atto565), composed of the N-terminus of CXCL11 and the main body and C-terminus of CXCL12 [32]. The latter was shown to be specifically bound and scavenged by ACKR3—but not by CXCL12's or CXCL11's natural receptors, i.e. CXCR4 or CXCR3 [32]. These experiments confirmed strong ACKR3-mediated scavenging activity in TNFα/IFNγ stimulated ndLECs at 37˚C (**Fig 1C and 1D**). Upon treatment with CCX771, CXCL11/12-Atto565 uptake was almost completely abrogated in TNFα/IFNγ stimulated ndLECs (**Fig 1C and 1D**). Similarly, also CXCL11/12 labeled with AlexaFluor647 (CXCL11/12-AF647) was taken up by TNFα/IFNγ stimulated, and to a lesser extent also by unstimulated ndLECs in an ACKR3-dependent manner (**Fig 1E and 1F**). To investigate whether AM competes with CXCL11/12 for ACKR3 binding, we next performed a ligand competition assay, in which we tested the ability of unlabeled, recombinant human AM to outcompete CXCL11/12-AF647 binding to ACKR3 and diminish its scavenging. To this end, ndLECs were again stimulated overnight in presence of TNFα/IFNγ and incubated for one hour with 50nM CXCL11/12-AF647 in presence of increasing concentrations of AM (0–10'000nM). While the uptake of CXCL12/12-AF647 (50nM) was efficiently blocked in presence of CCX771, even a 200-fold excess of AM failed to diminish CXCL11/12 scavenging (**Fig 1G**). Thus, AM was not able to out-compete the chemokine scavenging activity of ACKR3 in ndLECs.

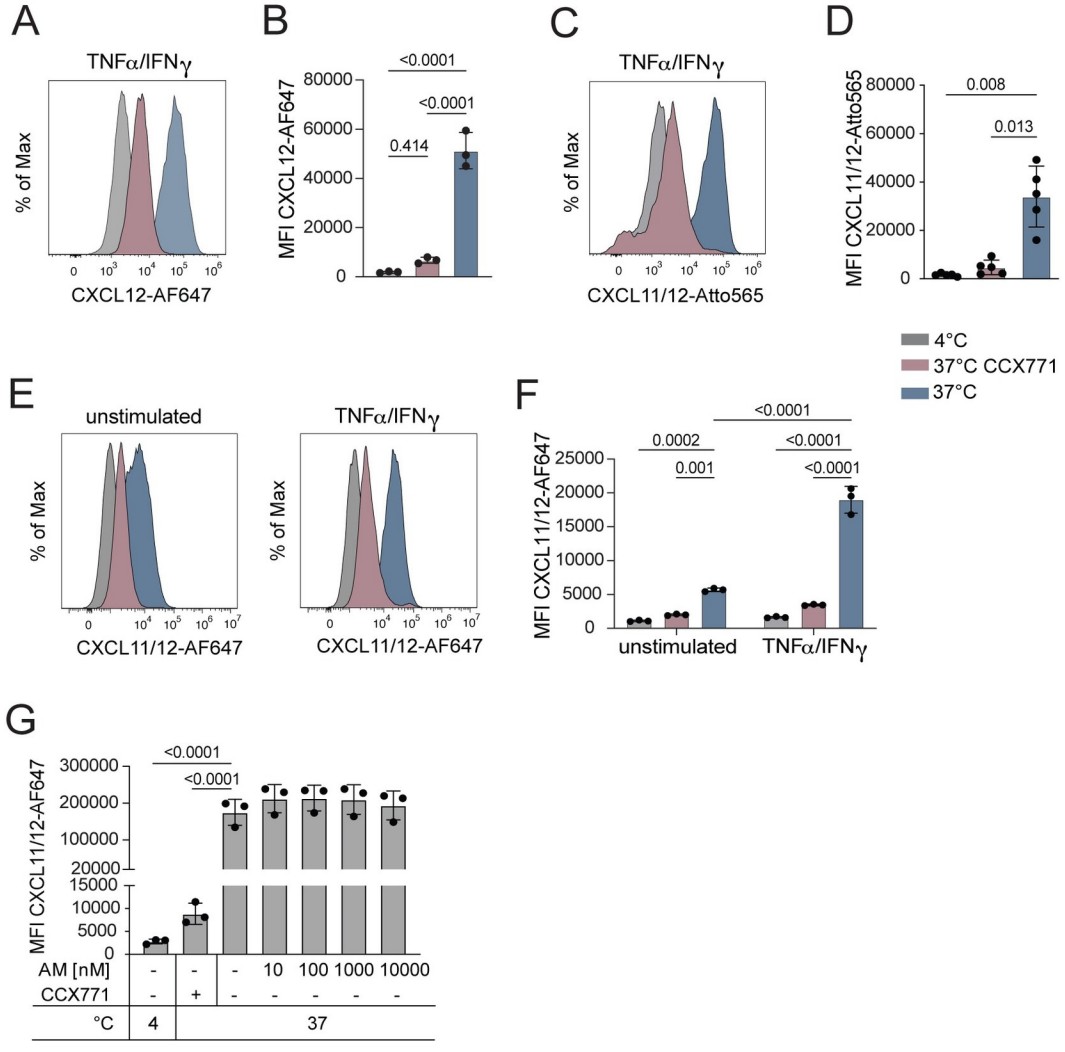

**Fig 1. Addition of AM does not reduce the chemokine scavenging activities of ACKR3 in a ligand competition assay in ndLECs *in vitro*.** (**A-D**) ndLECs stimulated by overnight treatment with TNFα/IFNγ take up (**A, B**) CXCL12-AF647 or (**C, D**) the ACKR3-specific chimeric chemokine CXCL11/12-Atto565 at 37˚C but not at 4˚C. Uptake at 37˚C is abrogated by treatment with CCX771. (**A, C**) Representative histograms and (**B, D**) quantifications of 3–5 independent experiments. One-way ANOVA. (**E, F**) Scavenging of the ACKR3-specific chimeric chemokine CXCL11/12-AF647 at 37˚C by unstimulated and TNFα/IFNγ stimulated human LECs is abrogated by treatment with CCX771 (**E**) Representative histograms and (**F**) quantification of 3 independent experiments. Two-way ANOVA. (**G**) AM is not able to displace CXCL11/12-AF647 in a ligand competition assay. CXCL11/12-AF647 (50nM) scavenging assay was performed in presence of exceeding concentrations of AM. Data of three independent experiments are shown as mean ±SD. Each dot represents the value obtained in an individual experiment. Statistics: RM One-way ANOVA, Dunnett's multiple comparison test.

## AM internalization in primary LECs is not altered by pharmacologic modulation of ACKR3

In order to visualize AM scavenging in LECs, we custom-synthesized an AF568-labelled AM molecule (AM-AF568) (**Fig 2A**). Similarly to unmodified AM [35], the chosen format had previously been shown to be internalized by HEK293 cells expressing the conventional AM receptor AM1, consisting of CALCRL and RAMP2 [31]. Notably, all LECs (ndLECs, jdLECs, adLECs) expressed CALCRL, RAMP2 and RAMP3 (**S1 Fig**). To assess the bioactivity of AM-AF568, we first tested its capacity to induce ndLEC proliferation *in vitro* [15, 36, 37]. In

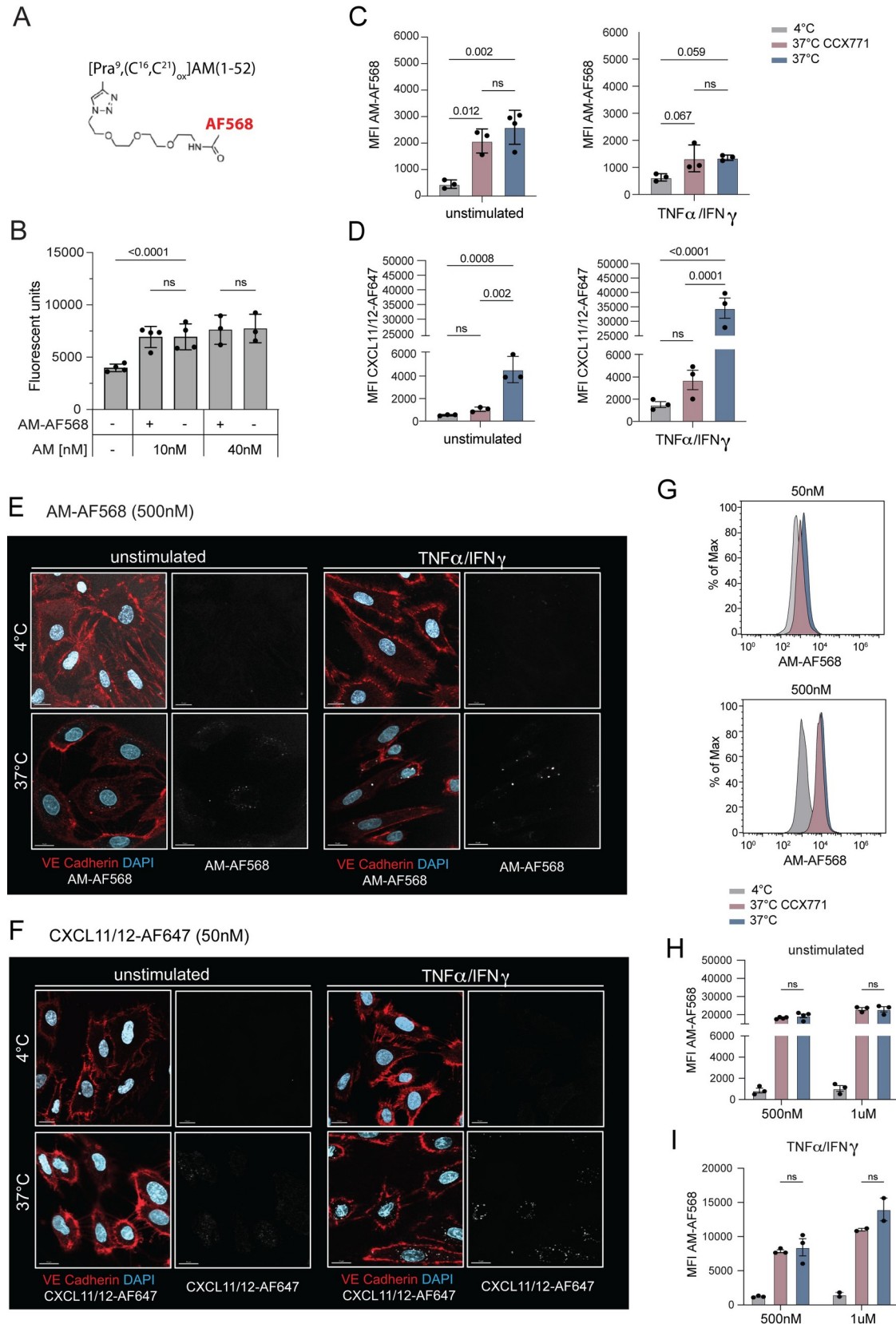

**Fig 2. AM scavenging in primary ndLECs is not reduced by pharmacologic manipulation of ACKR3.** (**A**) Structure of the custom-synthetized, disulfide bridged AM-AF568 according to [31]. The unnatural amino acid L-propargyl glycine (Pra) was inserted at position 9 of the AM sequence to allow for site-specific fluorescent labeling (see *Materials and methods*). (**B**) AM-AF568 induces comparable LEC proliferation as unlabeled AM. Each data point represents the mean of one experiment with 10 replicates. Mixed-effects analysis, Šídák's multiple comparison test. (**C, D**) Uptake of either 50nM AM-AF568 or 50nM CXCL11/12-AF568 by unstimulated or TNFα/IFNγ stimulated LECs in presence or absence of CCX771 was quantified by FACS. Effect of CCX771 on uptake of (**C**) AM-AF568 and (**D**) CXCL11/12-AF647. Pooled data from 3 experiments are shown. One-way ANOVA, followed by Tukey's multiple comparison test. (**E, F**) ndLECs were incubated with (**E**) AM-AF568 (500nM) or (**F**) CXCL11/12-AF647 (50nM) at either 4°C or 37°C and uptake into cells was analysed by confocal microscopy after 1h. Representative images from two independent experiments. Scale bars: 15μm. (**G-I**) LECs were incubated with either 50nM, 500nM or 1μM AM-AF568 in presence or absence of CCX771, and uptake was analysed by flow cytometry. (**G**) Representative histogram showing AM-AF568 internalization in stimulated LECs incubated with 50nM and 500nM. (**H, I**) Pooled data from three independent experiments performed with (**H**) unstimulated LECs and from two independent experiments performed with (**I**) TNFα/IFNγ stimulated LECs are shown. Results are shown as mean ±SD. Two-way ANOVA, followed by Tukey's multiple comparison test. A p-value of p≥0.05 was considered not significant (ns).

these experiments, synthesized AM-AF568 displayed comparable activity to commercial, unlabeled human AM in inducing LEC proliferation (**Fig 2B**). In agreement with the previous publication working with AM1-transfected HEK293 cells [31], AM-AF568 (50nM) was taken up at low levels by unstimulated ndLECs and by TNFα/IFNγ stimulated ndLECs (**Fig 2C**). Notably, the latter had downregulated the AM receptor components RAMP2 and RAMP3 (**S1 Fig**). However, in contrast to uptake of CXCL11/12-AF647 (**Fig 2D**), uptake of AM-AF568 was not reduced upon treatment of ndLECs with CCX771 (**Fig 2C**).

Considering that a recent study working with ACKR3-overexpressing cell lines only detected AM binding to ACKR3 at AM concentrations in the sub-micromolar range [9], we repeated the internalization experiment using 500nM and 1μM of AM-AF568 in unstimulated and TNFα/IFNγ stimulated ndLECs. Notably, at a concentration of 500nM AM-AF568 internalization was detected by confocal microscopy (**Fig 2E**). Similarly, internalization of CXCL11/12-AF647 could be observed, albeit at much lower concentrations (50nM) (**Fig 2F**). When analyzing uptake of AM-AF568 by FACS in either unstimulated or TNFα/IFNγ stimulated LECs, MFIs were strongly increased at 500nM and 1μM concentrations (**Fig 2G and 2H**), as compared to 50nM (**Fig 2C**). However, also at these high concentrations, CCX771 treatment did not reduce AM-AF566 scavenging (**Fig 2G and 2H**), suggesting that the cellular uptake observed by confocal microscopy (**Fig 2E** and [31]) was not dependent on ACKR3.

## ShRNA-mediated knockdown of ACKR3 does not impede AM internalization and uptake

To exclude the possibility that CCX771 and chemokines bind ACKR3 at a different site compared to AM and consequently might not compete with the AM scavenging function of ACKR3, we decided to knockdown ACKR3 expression in LECs by lentiviral transduction with shRNA. ACKR3 knockdown was performed in both adLECs and jdLECs using three different ACKR3-targeting shRNA-sequences (shRNA A-C). qRT-PCR-based analysis revealed that shRNA clones B and C effectively reduced *Ackr3* mRNA expression in unstimulated or TNFα/IFNγ stimulated LECs, while no reduction could be achieved with clone A (**S2 Fig**). The qRT-PCR-based results nicely correlated with the CXCL11/12-AF647 scavenging function, which was significantly reduced in TNFα/IFNγ stimulated adLECs treated with shRNAs B and C, but not in untreated adLECs (Ctrl) or adLECs treated with the ineffective shRNA clone A (**Fig 3A**). In contrast, when in parallel performing uptake experiments with AM-AF568, we did not detect any difference in the MFI between TNFα/IFNγ stimulated ACKR3-sufficient (i.e. Ctrl adLECs or adLECs treated with shRNA A) and ACKR3 knockdown adLECs (i.e.

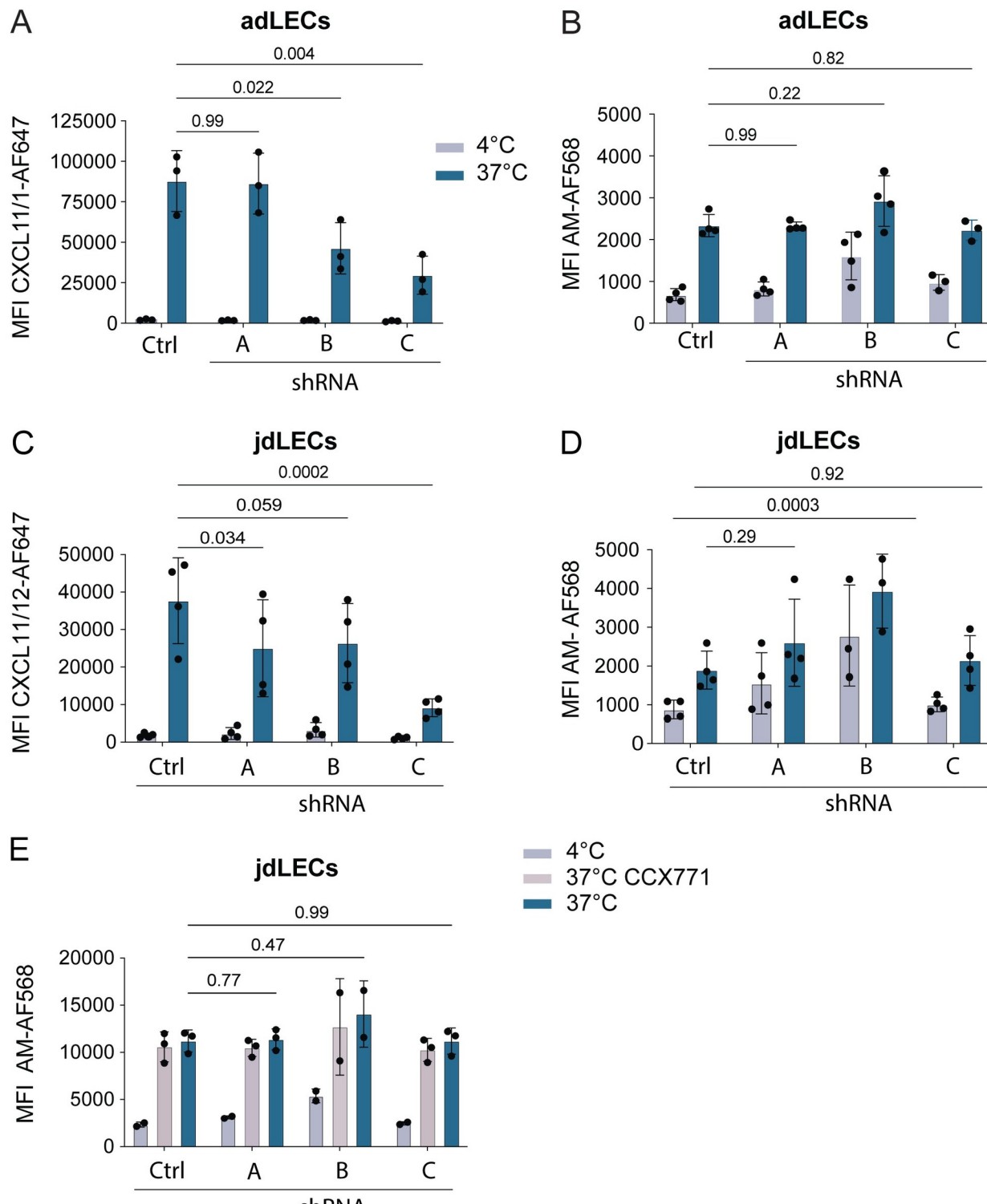

**Fig 3. shRNA-mediated ACKR3 knockdown does not diminish AM scavenging in TNFα/IFNγ stimulated primary adLECs or jdLECs.**
adLECs and jdLECs were transduced with ACKR3-specific shRNA constructs (A: no knockdown, B,C; 50–80% knockdown–see S3 Fig),
scrambled shRNA (shRNA Ctrl) or untransduced control LECs (Ctrl). Cells were stimulated by overnight incubation with TNFα/IFNγ.
Subsequently, cells were incubated with either CXCL11/12-AF647 (50nM) or AM-AF568 (50nM) and uptake analysed by flow cytometry. While
CXCL11/12-AF647 scavenging correlated with ACKR3 knockdown in (**A**) adLECs and (**C**) jdLECs, no impact of ACKR3 levels on AM-AF568
scavenging was observed in (**B**) adLECs and (**D**) jdLECs. (**E**) Also when performing the experiment with 500nM AM-AF568, no impact of

ACKR3-knockdown on scavenging was observed. Data from three or four independent experiments are shown as mean ±SD. Each data point represents one replicate. Two-way ANOVA, followed by Tukey's multiple comparison test (**A, C**). Mixed effects model, followed by Tukey's multiple comparison test, was applied, when repeated measures pairing was incomplete, due to subclone availability (**B, D, E**). A p-value of p≥0.05 was considered not significant (ns).

adLECs treated with shRNA clone B and C) (**Fig 3B**). Similarly, shRNA mediated knockdown of ACKR3 with shRNA clone C resulted in a significantly reduced uptake of CXCL11/12-AF647 but did not reduce AM-AF568 uptake in jdLECs (**Fig 3C and 3D**). Surprisingly, knockdown of ACKR3 with shRNA B resulted in a less robust reduction in CXCL11/12-AF647 uptake (**Fig 3C**), indicating that this knockdown was less robust as the one achieved with shRNA clone C. Since we could not detect any evidence of ACKR3-mediated scavenging when working with 50nM AM-AF568 (**Fig 3B and 3D**), we repeated the uptake experiments in TNFα/IFNγ stimulated jdLECs using a 10-fold higher concentration of 500nM AM-AF568. However, also at this concentration, no difference in the MFI of AM-568 uptake was observed between ACKR3-sufficient and ACKR3-deficient jdLECs (**Figs 3E and S3**).

To exclude any bias that might have been induced by the TNFα/IFNγ stimulation, we also repeated the side-by-side uptake experiments with CXCL11/12-AF647 and AM-AF568 in unstimulated shRNA-transduced adLECs and jdLECs, but also under these conditions no evidence for ACKR3 mediated AM scavenging was observed (**S4 Fig**). In summary, we did not observe any difference in AM-AF568 scavenging in ACKR3-shRNA silenced human LECs from two different sources as compared to ACKR3-sufficient controls. However, we observed that human LECs were able to take up residual amounts of AM-AF568 independently of ACKR3, likely by internalization mediated by canonical AM1 or AM2 receptors [31, 35].

## AM-mediated proliferation is not enhanced upon treatment with CCX771 or shRNA-mediated knockdown of ACKR3

According to the current model, LEC-expressed ACKR3 is thought to prevent overshooting responses of LECs towards AM. Consequently, one would expect stronger AM-induced responses to occur in absence of ACKR3, as a consequence of increased AM availability and signalling via AM1 (CALCRL:RAMP2) or AM2 (CALCRL:RAMP3) [14]. To investigate whether in our hands modulation of ACKR3 would impact LEC responsiveness towards AM (**Fig 4A**), we performed proliferation assays in presence of the ACKR3-selective competitive agonist CCX771 or upon shRNA-mediated knockdown of ACKR3. Treatment with CCX771 (1μM) by itself did not alter basal ndLEC proliferation (**Fig 4B**) and did not enhance LEC proliferation in response to AM concentrations ranging from 0.01 – 10nM (**Fig 4C**). Moreover, in contrast to published results [14], we did not observe that shRNA-mediated ACKR3 knockdown (clones B,C) enhanced adLEC proliferation in response to AM, in comparison to the proliferative response observed in control LECs (**Fig 4D and 4E**). Similarly, no increased AM responsiveness was observed in jdLECs in which ACKR3 had been knocked down by shRNA treatment (clones B,C) (**Fig 4F and 4G**). Thus, despite observing some differences and inter-assay variability in baseline cell growth between the different shRNA-treated LECs and control-treated cells (**Fig 4D and 4F**), the relative AM responsiveness of ACKR3-deficient was not increased as compared to ACKR3-sufficient LECs (**Fig 4E and 4G**). In summary, we did not observe that AM-induced proliferation was enhanced upon pharmacologic blockade or shRNA-mediated knockdown of ACKR3 in human LECs obtained from three different sources (i.e. ndLECs, jdLECs or adLECs).

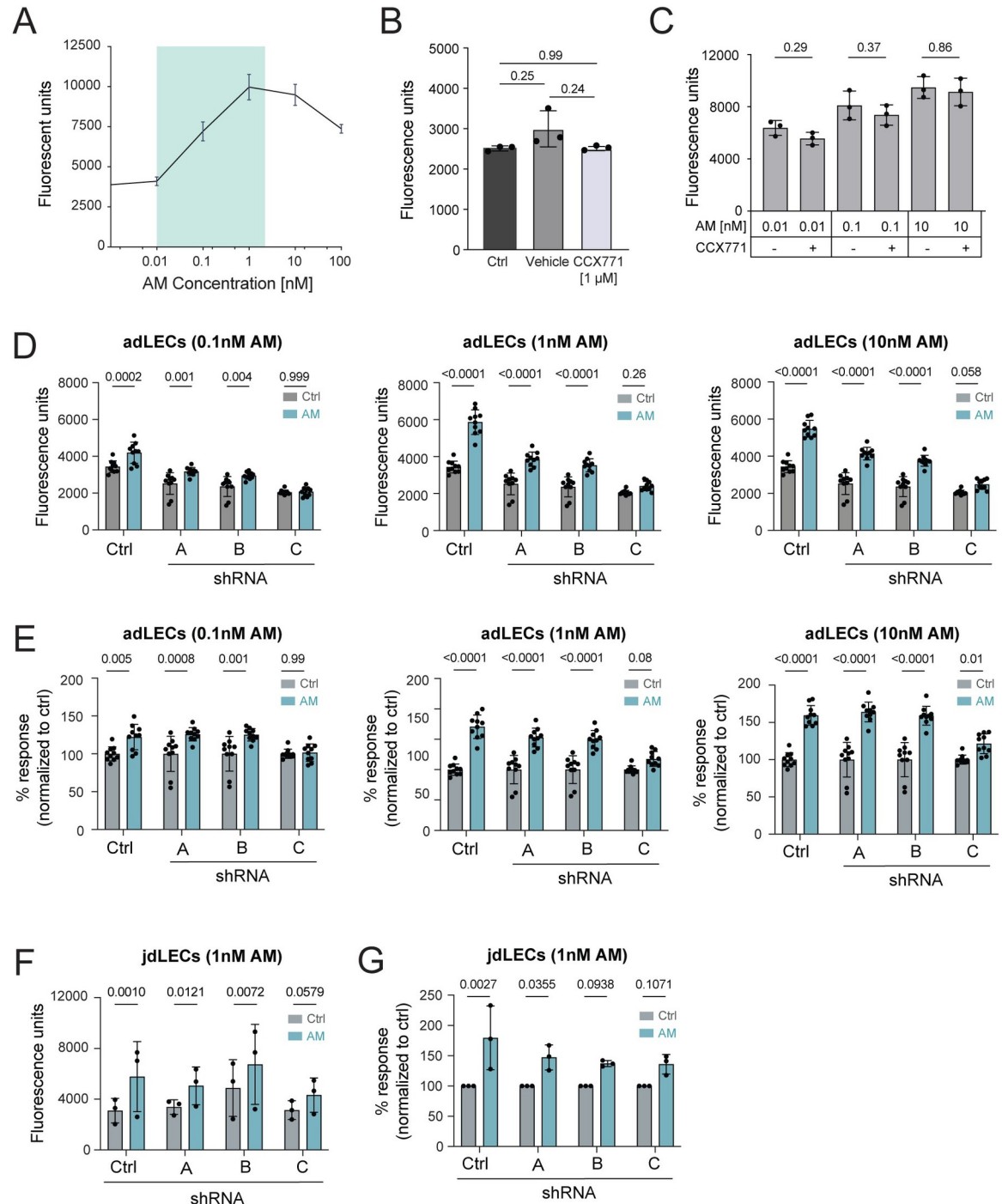

**Fig 4. AM-mediated proliferation is not enhanced upon treatment with CCX771 or shRNA-mediated knockdown of ACKR3.** (**A**) Titration of AM in a LEC proliferation assay. Data from one out of two similar titration assays, performed with ndLECs and adLECs with 10 technical replicates per condition, are shown. (**B**) Treatment with CCX771 does not affect baseline proliferation of ndLECs. (**C**) CCX771 treatment does not alter AM-induced proliferation of primary human ndLECs. Data of three experiments with 10 technical replicates are shown in (B,C) and represented as mean ±SD. RM One-way ANOVA, Šídák's multiple comparison test. (**D**) shRNA-mediated knockdown of ACKR3 does not alter proliferation induced by either 0.1nM, 1nM and 10nM AM in adLECs. Data of one out of one or two similar experiments with 10 technical replicates per condition are shown. (**E**) Data from (**D**) are shown as percentage of proliferation induced when compared to the corresponding untreated control. (**F, G**) Proliferation induced by 1nM AM was investigated in shRNA-transduced jdLECs. Data of three experiments are shown and represented as mean ±SD. Each data point represents the mean of 8 technical replicates, showing (**F**) Absolute fluorescence units as a readout of proliferation or (**G**) Percentage of

proliferation induction relative to the untreated control. Statistics: Two-way ANOVA, Šídák's multiple comparison test (D,E). RM-Two-way ANOVA, Šídák's multiple comparison test (F,G). A p-value of p≥0.05 was considered not significant (ns).

## AM-AF568 uptake in HEK293 cells exclusively depends on CALCRL and RAMP2/3 expression but not on ACKR3 expression

AM exerts its biological effects, such as induction of proliferation, by binding to its conventional receptors AM1 (CALCRL:RAMP2) and AM2 (CALCRL:RAMP3) [17]. Moreover, stimulation with AM was shown to induce AM internalization in complex with AM1 and AM2 [31, 35]. To exclude that in our primary LECs the ACKR3-dependency of AM-AF568 internalization was masked by stronger AM-AF568 binding/ internalization via LEC-expressed AM1 or AM2, we decided to test AM scavenging in a system with controlled CALCRL, RAMP2 or RAMP3 and ACKR3 expression. To this end, we made use of HEK293 cells stably transfected with ACKR3 bearing a hemagglutinin (HA) tag at its N-terminus (HEK293-ACKR3). ACKR3 expression was confirmed in HEK293-ACKR3 cells by flow cytometry (Fig 5A). Notably, neither the untransfected HEK293 control cells (ACKR3-CTRL) nor HEK293-ACKR3 cells expressed endogenous levels of *Ramp2* or *Calcrl*, as shown by qRT-PCR (S5 Fig). While stably transfected HEK293-ACKR3 cells scavenged CXCL11/12-AF647 (50nM) in comparison to HEK293-CTRL (Fig 5B and 5C), both cell types failed to scavenge AM-AF568 (50nM) (Fig 5D and 5E). We next transiently transfected both HEK293-CTRL and HEK293-ACKR3 cells with plasmids encoding CALCRL C-terminally fused to GFP (CALCRL-GFP) and RAMP2 N-terminally expressing a FLAG tag (RAMP2-FLAG) or a combination of both and subsequently performed AM-AF568 uptake experiments in these cell lines. Similarly, to the baseline HEK293-CTRL and HEK293-ACKR3 cells, neither RAMP2-FLAG nor CALCRL-GFP single-transfectants displayed any AM-AF568 internalization (Fig 5F). Conversely, in RAMP2-FLAG/ CALCRL-GFP double-transfected HEK293-CTRL and HEK293-ACKR3 cells, approximately 20% of all cells were positive for AM-AF568 (Fig 5F). Thus, the ability to internalize/ scavenge AM did not depend on the expression of ACKR3 but exclusively on expression of both AM1 receptor components (Fig 5F). Indeed, analysis by flow cytometry revealed that cells that had internalized AM-AF568 expressed RAMP2-FLAG on their surface and were, at the same time, positive for CALCRL-GFP (S5 Fig).

In addition to RAMP2 [36] also expression of RAMP3, which forms part of the AM2 receptor, was previously reported in isolated LECs and BECs [38, 39]. Moreover, RAMP3 has recently been implicated in ACKR3-mediated scavenging of AM [38]. Therefore, to investigate whether ACKR3's scavenging activity might depend on the expression of RAMP3, we also assessed AM scavenging in HEK293-CTRL and HEK293-ACKR3 cells transiently transfected with CALCRL-GFP, RAMP3-FLAG or both constructs. Similarly, to what we had seen in the case of RAMP2-FLAG transfectants (Fig 5F), only HEK293-CTRL and HEK293-ACKR3 cells double-transfected with CALCRL-GFP and RAMP3-FLAG were capable of internalizing AM-AF568 (Figs 5G and S5). Again, the efficiency of AM-AF568 uptake/ internalization did not depend on the presence of ACKR3 (Fig 5G).

In summary, in our hands, ACKR3 expression did not contribute to AM scavenging/ internalization, neither in primary human LECs nor in ACKR3-overexpressing HEK293 cells.

## Discussion

In this study, we further examined the prevailing concept that ACKR3 functions as scavenger of the vascular peptide hormone AM in LECs, thereby helping to prevent excessive AM-induced lymphangiogenic responses. To this end, we performed *in vitro* experiments in

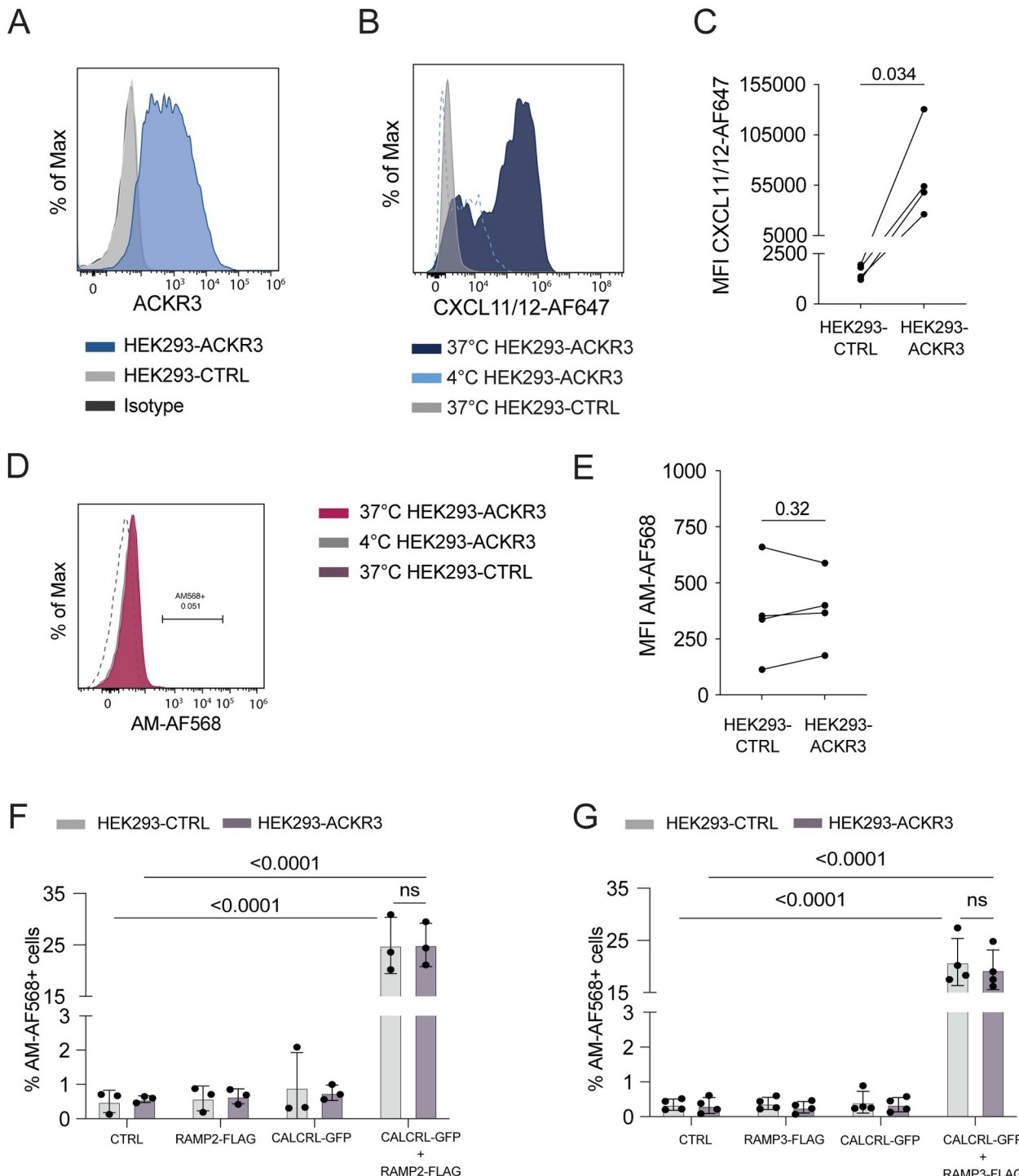

**Fig 5. AM scavenging in HEK293 cells requires co-expression of RAMP2/3 and CALCRL but is independent of ACKR3.** (**A**) ACKR3 is expressed in HEK293-ACKR3 but not in HEK293-CTRL cells. Antibody staining with anti-human ACKR3 (clone 11G8). (**B-E**) HEK293-ACKR3 or HEK293-CTRL cells were incubated with CXCL11/12-AF647 or AM-AF568 at either 4°C or 37°C and uptake activity was quantified by flow cytometry. While (**B, C**) CXCL11/12-AF647 was avidly scavenged by HEK293-ACKR3 cells, (**D, E**) no evidence of uptake was observed for AM-AF568 in HEK293-ACKR3 cells. (A,B,D) Representative Histograms. (C,E) Pooled measurements of 3–4 independent experiments are shown as mean ±SD. Statistics: Paired Student's t-test. (**F, G**) HEK293-ACKR3 or HEK293-CTRL cells were transiently transfected with CALCRL, RAMP2 or RAMP3 encoding plasmids and combinations thereof and AM-AF568 uptake was investigated by flow cytometry. AM-AF568 internalization was exclusively dependent on co-expression of CALCRL with (**F**) RAMP2 or (**G**) RAMP3, regardless of ACKR3 expression. (F,G) Pooled measurements from 4 independent experiments are shown as mean ±SD. RM two-way ANOVA , Šídáks multiple comparison test. A p-value of p≥0.05 was considered not significant (ns).

primary human LECs, which endogenously express ACKR3 and proliferate in response to AM, as well as in HEK293 cells transfected with combinations of ACKR3 and/ or components of the AM receptors AM1 and AM2. While our experiments provided ample evidence of proliferation and/or internalization induced by AM binding to canonical AM receptors, none of our readouts supported the conclusion that ACKR3 acts as a cell-intrinsic AM scavenger at concentrations at which AM-induced proliferation in LECs is observed.

Whether or not ACKR3 binds and scavenges AM has been a topic of recurrent debate over the past years. Initial evidence had been gathered more than 20 years ago, when AM stimulation of the orphan canine RDC-1 receptor (now known as ACKR3) over-expressed in COS-7 cells was found to elicit a dose-dependent cAMP response with an $EC_{50}$ of 100nM [2]. Although several other putative AM receptors were brought forward at the time [40], AM1 and AM2, consisting of CALCRL in complex with either RAMP2 or RAMP3, were subsequently discovered as high affinity receptors of human AM, with reported binding affinities in the low nanomolar range [41]. New evidence for the receptor-ligand relationship of AM and ACKR3 was only published in 2014, when Klein et al. proposed ACKR3 as an AM scavenger, based on the striking phenotypic interrelationship between ACKR3-deficient and AM-overexpressing mouse embryos and overshooting *in vitro* responses of ACKR-deficient LECs towards AM [14]. In further support of this concept, two recent studies reported that stimulation with micromolar concentrations of AM induced β-arrestin recruitment to ACKR3 in ACKR3-overexpressing HEK293, indicative of binding but not necessarily scavenging [24, 38]. On the other hand, two additional studies found that AM was not capable of displacing fluorescent CXCL12 from ACKR3 in ligand competition assays, performed with either ACKR3-overexpressing U87 or HEK293 cells [23, 24]. In agreement with the latter results, the ligand competition assay performed in our study with primary human LECs also revealed that CXCL11/12-AF647 scavenging was not diminished in presence of a 200-fold excess amount of AM (10μM). Furthermore, we could not identify any evidence of ACKR3-dependent AM scavenging in human LECs when performing experiments at AM concentrations ranging from 50nM– 1μM.

Considering that β-arrestin recruitment was only observed in response to micromolar AM concentrations [24, 38], we cannot rule out that ACKR3-mediated AM scavenging in LECs might only be observable at even higher concentrations of AM than the ones used in our assays (up to 1μM). However, as previously reported, AM induces LEC proliferation, migration or ERK phosphorylation in LECs at optimal concentrations ranging from of 0.1 – 100nM [14, 15], as also seen in our experiments, where LEC proliferation was induced in presence of as little as 0.1nM AM (**Fig 4A**).Therefore, the AM concentrations triggering cellular responses in LECs appear to be at least two log units lower than the concentration at which AM binding to ACKR3 reportedly occurs [24, 38]. This makes it rather unlikely that ACKR3 might directly function as a physiological cell-intrinsic scavenger, regulating bioavailability of AM towards LEC-expressed AM receptors. In fact, in order to function as competitive cell-intrinsic scavenger, a receptor would be expected to display comparable if not increased affinity for its ligand in comparison to the conventional receptor competing for ligand binding. In agreement with this basic principle, ACKR3 exhibits comparable or higher affinity for its chemokine ligands CXCL11 and CXCL12 in comparison to the chemokines' affinity towards their conventional chemokine receptors CXCR3 and CXCR4, respectively [5, 42]. In light of these considerations, it seems perhaps not so surprising that we could not observe any evidence of enhanced LEC proliferation towards AM (0.1 – 10nM tested) upon pharmacologic inhibition of ACKR3 or lentiviral knockdown of ACKR3 expression. Although we consistently obtained the same results in experiments performed with different shRNAs and LEC cell types (adLECs and jdLECs, **Fig 4A–4D**), we cannot exclude that the reason why different results, namely, enhanced AM-

induced proliferation upon shRNA-mediated knockdown of ACKR3, were observed by Klein et al. [14] might lie in the different cellular source and shRNA clone used in these experiments.

Nonetheless, it remains possible that ACKR3 might indirectly affect AM responses mediated by conventional AM receptors, by e.g. competing with the binding of accessory molecules such as β-arrestins and G-protein-coupled receptor kinases (GRKs) required for optimal AM signaling and internalization. Intriguingly, overexpression studies in HEK293 cells have recently revealed that RAMP2 and RAMP3 associate with many chemokine receptors, including ACKR3 [38]. Specifically, association of RAMP3 with ACKR3 was shown to influence vesicular trafficking of ACKR3 following AM-induced internalization and rapid receptor recycling and re-sensitization [38]. On the other hand, our experiments performed in HEK293-CTRL or HEK293-ACKR3 cells revealed that internalization of AM-AF568 exclusively depended on the expression of CALCRL together with either RAMP2 or RAMP3, but was completely independent of ACKR3 expression. Notably, these findings are in line with Meyrath et al, who reported that AM-induced β-arrestin recruitment was not altered by co-expression of RAMPs together with ACKR3 in HEK293 cells [9].

Our current findings indicating that ACKR3 displays no—or at best poor–cell-intrinsic AM scavenging activity in LECs is further supported by a recent study from our lab showing that conditional knockdown of ACKR3 in LECs in newborn mouse pups did not impact post-natal lymphatic development and lymphatic drainage [25]. Despite the findings from our previous [25] and the present study, it remains possible that the impressive lymphatic hyperproliferation and edema phenotype observed in global ACKR3 knockout embryos, which lack ACKR3 expression not only in LECs but in many other cell types, is in some way caused by altered signalling induced by AM, its receptors or potentially even by related, AM-derived vasoactive peptides: of interest in this context, PAMP-12, a derivative of the proadrenomedullin N-terminal 20 peptide (PAMP), was recently found to bind ACKR3 with nanomolar affinity, inducing β-arrestin recruitment and receptor internalization in ACKR3-expressing HEK293 cells [24]. Notably, AM and PAMPs derive from the same pro-adrenomedullin precursor [43]. Although a G-protein coupled receptor (RMGPRX2 [44]) has been identified as a receptor of PAMPs including PAMP12, the physiologic functions of PAMP12 remain largely unknown [24]. The fact that both AM and PAMP-12 are encoded by the adrenomedullin locus adds an additional layer of complexity to the phenotypic observations made in *in vivo* models of AM knockout or overexpression. It will therefore remain important to further dissect the complex relationship between ACKR3, AM receptors and pro-adrenomedullin derivatives, to hopefully better understand the striking phenotypic similarity observed between ACKR3-deficient and AM-overexpressing mouse embryos [14].

## Supporting information

**S1 Fig. Characterization of LECs used in experiments.** (**A**) FACS analysis showing expression of CD31 and podoplanin in ndLECs, jdLECs and adLECs. (**B**) qRT-PCR was performed to investigate *Ackr3* mRNA expression in steady-state (unstim) and TNFα/ IFNⓇ stimulated (stim) ndLECs, jdLECs and adLECs. *Prox-1* levels were determined for comparison. CT values are shown on the left and resulting fold-changes on the right. (**C-F**) ACKR3 protein expression was not detectable in resting ndLECs (unstim) by flow cytometry, but could be detected on the ndLEC cell surface after TNFα/ IFNγstimulation (stim). Staining was performed with two different anti-ACKR3 antibodies, i.e. with (**C, D**) clone 9C4 and (**E, F**) clone 11G8. Representative histograms are shown in (C,E) and quantifications of 3–4 independent experiments in (D, E). Each dot represents the ΔMFI value (normalized to the isotype control) from one staining. Student's t-test. (**G**) qRT-PCR was performed to investigate *Calcrl*, *Ramp2* and *Ramp3* mRNA

expression in steady-state (Ctrl) and TNFα/ IFN® stimulated (stim) ndLECs, jdLECs and adLECs. Data from 1 RNA extraction and 3 technical replicates are shown in (B,G). Statistical analysis performed in (B,G): One sample t test.
(TIF)

**S2 Fig. qRT-PCR-based characterization of shRNA-transduced adLECs and jdLECs.** Validation of shRNA-mediated ACKR3 knockdown in lentivirally transduced and subsequently sorted adLECs and jdLECs, in comparison to untransduced (Ctrl) or scrambled RNA-transduced (shRNA Ctrl) cells. (**A**) Knockdown efficiency in steady-state adLECs was analysed at p6, shortly, after sorting and p9, after all experiments were performed. (**B**) Knockdown efficiency in resting jdLECs at p8. (**C**) Knockdown efficiency in TNFα/IFN® stimulated shRNA-transduced jdLECs at p8. Each data point represents the normalized relative expression calculated from three technical replicates per repetition.
(TIF)

**S3 Fig. Knockdown of ACKR3 in jdLECs prevents CXCL11/12-AF647 scavenging but does not impact AM-AF568 internalization.** Sorted jdLECs (p7), transduced with either scrambled control shRNA or ACKR3-targeting scRNA construct C, were first starved, then stimulated for 24h with 20ng TNFα/ IFN® or left untreated. Uptake assays were performed with AM-AF568 (500nM) or CXCL11/12-AF647 (50 nM). (**A, B**) Representative images of (**A**) CXCL11/12-AF647 and (**B**) AM-AF568 uptake performed in unstimulated (Ctrl) jdLECs. (**C, D**) Representative images of 2 uptake experiments performed with (**C**) CXCL11/12-AF647 and (**D**) AM-AF568 in TNFα/IFNγ stimulated jdLECs. Scale bars: 10μm.
(TIF)

**S4 Fig. shRNA-mediated knockdown of ACKR3 does not diminish AM uptake in unstimulated primary adLECs or jdLECs.** adLECs and jdLECs were transduced with ACKR3-specific shRNA constructs (A: no knockdown, B,C; 50–80% knockdown–see S2 Fig), scrambled shRNA (shRNA Ctrl) or untransduced control LECs (Ctrl). For uptake assays, cells were incubated with either CXCL11/12-AF647 (50nM) or AM-AF568 (50nM) and uptake analysed by FACS. While CXCL11/12-AF647 scavenging correlated with ACKR3 knockdown in (**A**) adLECs and (**C**) jdLECs, no impact of ACKR3 levels on AM-AF568 scavenging was observed in (**B**) adLECs and (**D**) jdLECs. (**E**) Also, when performing the experiment with 500nM AM-AF568, no impact of ACKR3-knockdown on scavenging was observed. Data from three or four independent experiments are shown as mean ±SD. Each data point represents one replicate. Statistics: Two-way ANOVA, followed by Tukey's multiple comparison test (**C**). A mixed effects model, followed by Tukey's multiple comparison test, was applied in all other cases, due to incomplete repeated measures-pairing (**A, B, D, E**). A p-value of p≥0.05 was considered not significant (ns).
(TIF)

**S5 Fig. Validation of the HEK293 overexpression system.** (**A, B**) Analysis of mRNA expression in untransfected HEK293 control (HEK293-CTRL) and HEK293-ACKR3 cells indicated no endogenous expression of *Ackr3*, *Calcrl* or *Ramp2* in HEK293-CTRL cells. (**A**) Comparative mRNA expression between HEK293-CTRL and HEK293-ACKR3 cells. Data are shown as mean ±SD. (**B**) Fold change. Values from one qRT-PCR experiment, with three technical replicates are shown. (**C, D**) FACS plots of uptake experiments performed in presence of 50nM AM-AF568 in HEK293-CTRL and HEK293-ACKR3 cells transiently transfected with (**C**) CALCLRL-GFP and RAMP2-FLAG or with (**D**) CALCLRL-GFP and RAMP3-FLAG. Back-gating (dot plot on right) demonstrated that AM-AF568 was exclusively scavenged by cells co-expressing both constructs. Representative Dot plots of one out of four independent

experiments are shown in (**C, D**).
(TIF)

## Acknowledgments

The authors thank Thomas Schall (ChemoCentryx Inc., Mountain View, CA, USA) for providing CCX771. Moreover, they thank Angela Vallone, Lilian Baur and Katharina Blatter for excellent technical assistance.

## Author Contributions

**Conceptualization:** Elena C. Sigmund, Daniel F. Legler, Cornelia Halin.

**Data curation:** Elena C. Sigmund.

**Formal analysis:** Elena C. Sigmund, Aline Bauer, Hazal Tatliadim.

**Funding acquisition:** Marcus Thelen, Daniel F. Legler, Cornelia Halin.

**Investigation:** Elena C. Sigmund, Aline Bauer, Barbara D. Jakobs, Hazal Tatliadim, Carlotta Tacconi, Marcus Thelen.

**Methodology:** Barbara D. Jakobs, Carlotta Tacconi, Marcus Thelen, Daniel F. Legler.

**Resources:** Cornelia Halin.

**Supervision:** Cornelia Halin.

**Visualization:** Elena C. Sigmund, Aline Bauer, Hazal Tatliadim.

**Writing – original draft:** Elena C. Sigmund, Cornelia Halin.

**Writing – review & editing:** Elena C. Sigmund, Barbara D. Jakobs, Marcus Thelen, Daniel F. Legler, Cornelia Halin.

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
