## [Decision Letter · Decision Letter 0]

13 Feb 2023

PONE-D-23-00791

Reassessing the adrenomedullin scavenging function of ACKR3 in lymphatic endothelial cells

PLOS ONE

Dear Dr. Halin,

Thank you for submitting your manuscript to PLOS ONE. After careful consideration, we feel that it has merit but does not fully meet PLOS ONE’s publication criteria as it currently stands. Therefore, we invite you to submit a revised version of the manuscript that addresses the points raised during the review process.

One of the two reviewers asks for a minor revision and the second for a major revision, particularly on the part on the part concerning the experimental procedures followed for the construction and transfection of the plasmid.

We look forward to receiving your revised manuscript.

Kind regards,

Gabriella Lupo

Academic Editor

PLOS ONE

Journal Requirements:

“C.H., M.T. and D.F. Legler gratefully acknowledge common financial support from the Swiss National Science Foundation (https://www.snf.ch/en) Sinergia program (CRSII3_160719 / 1) and C.H additional support (core funding) from ETH Zurich.”

“The authors thank Thomas Schall (ChemoCentryx Inc., Mountain View, CA, USA) for providing CCX771. Moreover, they thank Angela Vallone, Lilian Baur and Katharina Blatter for excellent technical assistance. C.H., M.T. and D.F. Legler gratefully acknowledge common financial support from the Swiss National Science Foundation (https://www.snf.ch/en) Sinergia program (CRSII3_160719 / 1) and C.H additional support from ETH Zurich.”

“C.H., M.T. and D.F. Legler gratefully acknowledge common financial support from the Swiss National Science Foundation (https://www.snf.ch/en) Sinergia program (CRSII3_160719 / 1) and C.H additional support (core funding) from ETH Zurich.”

Reviewers' comments:

Reviewer's Responses to Questions

**Comments to the Author**

1. Is the manuscript technically sound, and do the data support the conclusions?

Reviewer #1: Partly

Reviewer #2: Yes

2. Has the statistical analysis been performed appropriately and rigorously? 

Reviewer #1: I Don't Know

Reviewer #2: Yes

3. Have the authors made all data underlying the findings in their manuscript fully available?

Reviewer #1: Yes

Reviewer #2: Yes

4. Is the manuscript presented in an intelligible fashion and written in standard English?

Reviewer #1: Yes

Reviewer #2: Yes

5. Review Comments to the Author

Reviewer #1: This manuscript from the Halin laboratory has reassessed the established notion (Klein et al. Dev Cell 2014) that ACKR3 acts as a scavenger of the opioid peptide adrenomedullin (AM) in LECs. They have convincingly demonstrated that, in their hands, ACKR3 does not have scavenging activity on AM at physiologically relevant concentrations either in primary LECs (from multiple sources) or transfected cell lines. They have used a combination of approaches to demonstrate this, including knock down of ACKR3 and an ACKR3-specific inhibitor. The experiments are generally well controlled and unequivocal. The manuscript is well-written, appropriately discussed and the data generally are well presented. There are a few concerns regarding statistical analysis, rigour and data interpretation that should be addressed prior to publication however.

1) Figure 1A: data is from a single experiment. This should be done more than once at a minimum.

2) Figure 1A and 1B: quantification and statistical analysis should be provided here.

3) Figure 2C: a direct statistical comparison between the MFI of the '37C AM-AF568' condition between the unstim and TNFa/IFNg groups should be provided to support the claim made on line 354 of the results.

4) Figure 4E and 4F: To show that the signal measured for AM/chemokine uptake in these assays is specific, a negative control should be provided (e.g. cells at 4C)

5) Figure 4H and 4I: the 50nM group depicted in these plots appears to be identical to the data shown in Figure 2C. This group should be removed from 4H and 4I to avoid this duplication.

6) It is notable that the data shown in Figure 4D and 4E appear to suggest that the KD of ACKR3 using shRNA 'C' has actually inhibited AM-induced LEC proliferation. While this is clearly the opposite of what would be expected based on the data published by Klein et al. Dev Cell 2014, the authors should discuss what might underly this.

7) Some of the p values being reported in the manuscript do not seem to match the distribution of data points as presented. Figure 4F (Ctrl: Ctrl v AM p=0.0010); Supp Figure 4C (Ctrl 37C v shRNA C 37C p=0.0004). Please clarify how these values have been computed.

8) Statistical tests and p values should be provided in Supp Fig 1B and SF1G to support claims made in the manuscript.

Minor: human gene names should be italicised all caps not lower case for annotation of genes in qPCR

Reviewer #2: In the study by Halin et Al., “Reassessing the adrenomedullin scavenging function of ACKR3 in lymphatic endothelial cells” the role of ACKR3 as scavenger of the vascular peptide hormone adrenomedullin (AM) was clarified. In this work, in vitro experiments were performed to evaluate the uptake of AM in primary human Lymphatic Endothelial Cells (LECs) and in ACKR3-overexpressing Human Epithelial Kidney (HEK) cells. The aim of this work was to verify a competition between the normal ACKR3-ligand CXCL11/12 and AM, to evaluate if this receptor can influence/reduce the AM proliferative effects on LECs. The Authors conclude that the AM does not compete with CXCL11/12 for ACKR3 binding, but it binds their canonical receptor (CALCRL and PAMP2/3. In my opinion this finding may have a relevance in this field, but major revisions are required.

The background should be improved, experimental procedures should be better described and rearranged. For example, first cell culture (including HEK cells), second the treatments and silencing on LECs, then the plasmid construction and transfection, finally the assay.

A few questions:

Why three type of LECs were used?

Why the coating was different between adLECs and nd- and jd-LECs?

The abbreviations should be uniformed and the same name should be used for the same protein, for example: ACKR3 was used throughout manuscript, but to indicate the antibody against this protein the CXCR/RDC-1 APC was used. The terms CXCL11/12-AF647 should be clarified as extended terms of CXCL; AF647 should be described as alexa flour 647; Atto 565 should be explained.

In the introduction, CCX771 should be described as an antagonist of ACKR3.

The description of the plasmids in the section “construction of expression plasmids for HEK293 cell transfection” should match with the description in the section “HEK293 cell transfection…”

In Figure1D, the error bars should be indicated in the graphic, which could be expanded to avoid the overlap between 37 °C ° and 37 °C + CCX771.

Please check and harmonize all the figures. For example, in fig 2D CXCL11/12 is reported as CXCl11/12; cell types are indicated in fig 3A, but not in fig 2.

6. PLOS authors have the option to publish the peer review history of their article (what does this mean?). If published, this will include your full peer review and any attached files.

Reviewer #1: No

Reviewer #2: No

---

## [Author Response · Author response to Decision Letter 0]

20 Apr 2023

and https://journals.plos.org/plosone/s/file?id=ba62/PLOSOne_formatting_sample_title_authors_affiliations.pdf

Response #1: We have updated the file names to fit PLOS ONE's style requirements.

“C.H., M.T. and D.F. Legler gratefully acknowledge common financial support from the Swiss National Science Foundation (https://www.snf.ch/en) Sinergia program (CRSII3_160719 / 1) and C.H additional support (core funding) from ETH Zurich.”

If this statement is not correct you must amend it as needed. Please include this amended Role of Funder statement in your cover letter; we will change the online submission form on your behalf.

Response #2: The funders had not role in the study design. Thus, please include the suggested statement: "The funders had no role in study design, data collection and analysis, decision to publish, or preparation of the manuscript."

Important: If there are ethical or legal restrictions to sharing your data publicly, please explain these restrictions in detail. Please see our guidelines for more information on what we consider unacceptable restrictions to publicly sharing data: http://journals.plos.org/plosone/s/data-availability#loc-unacceptable-data-access-restrictions. Note that it is not acceptable for the authors to be the sole named individuals responsible for ensuring data access

Response #3: The data presented in this study will be made openly available in the ETH Research Collection (https://www.research-collection.ethz.ch/) upon acceptance. At this point, we will be able to generate and receive a DOI. 

Response #4: The data presented in this study will be made openly available in the ETH Research Collection (https://www.research-collection.ethz.ch/) upon acceptance. At this point, we will be able to generate and receive a DOI. 

“The authors thank Thomas Schall (ChemoCentryx Inc., Mountain View, CA, USA) for providing CCX771. Moreover, they thank Angela Vallone, Lilian Baur and Katharina Blatter for excellent technical assistance. C.H., M.T. and D.F. Legler gratefully acknowledge common financial support from the Swiss National Science Foundation (https://www.snf.ch/en) Sinergia program (CRSII3_160719 / 1) and C.H additional support from ETH Zurich.”

Response #5: We have removed the funding statement from the Acknowledgement section. The latter now reads as follows: 

“The authors thank Thomas Schall (ChemoCentryx Inc., Mountain View, CA, USA) for providing CCX771. Moreover, they thank Angela Vallone, Lilian Baur and Katharina Blatter for excellent technical assistance.”

“C.H., M.T. and D.F. Legler gratefully acknowledge common financial support from the Swiss National Science Foundation (https://www.snf.ch/en) Sinergia program (CRSII3_160719 / 1) and C.H additional support (core funding) from ETH Zurich.”

Response #5: The funding statement should remain unchanged: 

“C.H., M.T. and D.F. Legler gratefully acknowledge common financial support from the Swiss National Science Foundation (https://www.snf.ch/en) Sinergia program (CRSII3_160719 / 1) and C.H additional support (core funding) from ETH Zurich.”

 

Comments from the Reviewers: 

Response to comments from Reviewer 1:

Reviewer #1: This manuscript from the Halin laboratory has reassessed the established notion (Klein et al. Dev Cell 2014) that ACKR3 acts as a scavenger of the opioid peptide adrenomedullin (AM) in LECs. They have convincingly demonstrated that, in their hands, ACKR3 does not have scavenging activity on AM at physiologically relevant concentrations either in primary LECs (from multiple sources) or transfected cell lines. They have used a combination of approaches to demonstrate this, including knock down of ACKR3 and an ACKR3-specific inhibitor. The experiments are generally well controlled and unequivocal. The manuscript is well-written, appropriately discussed and the data generally are well presented. There are a few concerns regarding statistical analysis, rigor and data interpretation that should be addressed prior to publication however.

1) Figure 1A: data is from a single experiment. This should be done more than once at a minimum.

Response: Sparked by the comment of the Reviewer, we have repeated the uptake experiments with CXCL12-AF647 two more times and now also provide quantification of the MFIs measured in all three experiments (Figure 1B), in addition to showing the representative histogram (Figure 1A). 

2) Figure 1A and 1B: quantification and statistical analysis should be provided here.

Response: We have also performed further repetitions of the uptake experiment with CXCL11/12-AF647 (previously shown in Figure 1B – now shown in Figure 1E) and provide the requested quantifications of this experiment in Figure 1F (and of Figure 1A, as discussed above). Notably, in the case of the uptake of CXCL11/12-AF647, we decided to repeat the experiment with more experimental groups than previously shown; namely, uptake performed in presence / absence of the inhibitor CCX771 in either unstimulated or stimulated LECs, in addition to the respective 4°C control. By including all 6 groups we are now able to provide statistical proof for our claim that ACKR3-mediated uptake is enhanced in TNF�/IFN�-stimulated LECs (Figure 1E,F – see also text in lines 360-362).

3) Figure 2C: a direct statistical comparison between the MFI of the '37C AM-AF568' condition between the unstim and TNFa/IFNg groups should be provided to support the claim made on line 354 of the results.

Response: Since – in contrast to the experiments quantified in Figure 1E,F - the uptake experiments shown in Figures 2C and 2D were not performed on the same day in unstimulated vs. stimulated LECs, we decided against pooling the data into one single graph and statistically comparing the stimulated and unstimulated condition. Consequently, we cannot claim that TNFa/IFNg-stimulated LECs take up less AM-AF568 as compared to unstimulated LECs. Sparked by the comment of the Reviewer, we have now rephrased the sentences describing the results of Figure 2C (lines 395-397). 

4) Figure 4E and 4F: To show that the signal measured for AM/chemokine uptake in these assays is specific, a negative control should be provided (e.g. cells at 4C)

Response: After consulting with our editor Dr. Lupo, we believe that the Reviewer was referring to the microscopy images shown in Figure 2E and 2F. We have added the requested negative control images from uptake experiments performed at 4°C in Figures 2E and 2F. Notably, this condition had been included in the original experiments and images had been acquired, but we had in the preparation phase of the manuscript decided against including the images in the Figure, for space reasons. The Figure legend has been adapted accordingly (lines 408-409). 

5) Figure 4H and 4I: the 50nM group depicted in these plots appears to be identical to the data shown in Figure 2C. This group should be removed from 4H and 4I to avoid this duplication.

Response: (We again assumed that the Reviewer was referring to Figures 2H and 2I). We thank the Reviewer for spotting this! The 50nM groups have now been removed from Figures 2H and 2I.

6) It is notable that the data shown in Figure 4D and 4E appear to suggest that the KD of ACKR3 using shRNA 'C' has actually inhibited AM-induced LEC proliferation. While this is clearly the opposite of what would be expected based on the data published by Klein et al. Dev Cell 2014, the authors should discuss what might underly this.

Response: We thank the Reviewer for spotting this! – Indeed, in Figure 4D/E LECs transduced with shRNA clone C, which induced the most effective knockdown of ACKR3, seemed to have a reduced proliferative response as compared to the other LEC clones. Although this was consistently observed at all AM concentrations in all experiments performed with adLECs (Figure 4D/F and data not shown), it is worth noticing that we did not see the same striking trend when performing experiments in shRNA-transduced jdLECs (Figure 4F/G). At this point, we therefore suspect that the differences observed could be due to different LECs sources - in our experiments but also in the study of Klein et al. – or linked with this particular shRNA clone. We therefore cannot conclude anything about a potential inhibitory effect of ACKR3 knockdown. However, we can say with certainty that - at least in the two cell types and the shRNAs we worked with - ACKR3 knockdown did not enhance proliferative responses towards AM, in contrast to what was reported by Klein et al. We have now inserted a sentence in the discussion speculating on why different results might have been reported by Klein et al (lines 621 - 625). 

While intensively discussing the data in Figure 4 E-G, we noticed that we had by mistake inserted a wrong panel in Figure 4D (left panel - 0.1nM AM), which did not correspond to the correct panel for the % response data shown in Figure 4E. In fact, the experimental data originally shown in Figure 4D (left panel – 0.1nM AM) belonged to a repeat experiment performed for all concentrations (0.1nM, 1.0 nM, 10 nM AM). This mistake has now been corrected and the panel (Figure 4D, left panel - 0.1nM AM) exchanged. As the Reviewer will see, the message remains the same; i.e. no enhancement of proliferation upon ACKR3 knockdown – potential loss of proliferative response with shRNA clone C. 

7) Some of the p values being reported in the manuscript do not seem to match the distribution of data points as presented. Figure 4F (Ctrl: Ctrl v AM p=0.0010); Supp Figure 4C (Ctrl 37C v shRNA C 37C p=0.0004). Please clarify how these values have been computed.

Response: Sparked by the comment of the Reviewer, we have once again checked the statistical analyses performed for Figures 4F and for Supplemental Figure 4C. Indeed, due to the different types of experiments reported in these two Figures, different types of comparisons and hence statistical analyses were performed: 

Figure S4C shows data from a chemokine uptake experiment, analyzed by flow cytometry. The statistical analysis performed in this case was a Repeated measures two-way ANOVA, followed by Tukey’s multiple comparison test, with a single pooled variance, for individual pairwise comparisons. Tukey’s multiple comparisons test was chosen because it compares between the means of all different groups. In this experiment, pairwise comparisons were made to assess the effect of different treatments (4°C binding ctrl, 37°C + CCX771, 37°C) on chemokine uptake, in relation to treatment with the different shRNAs (4 groups). Hence, pairwise comparisons for all different shRNA knockdown groups and conditions (4°C binding ctrl, 37°C + CCX771, 37°C) were calculated. Repeated-measures ANOVA was used due to the paired nature of the data, which came from different uptake experiments always performed with all different treatment and shRNA knockdown groups. However, for overview / clarity reasons, only the statistics of the relevant comparisons (i.e CTRL vs shRNA at 37°C) were displayed in the final Figure S4C.

Conversely, Figure 4F shows data from a proliferation assay, performed to assess AM-induced proliferation in the different shRNA-transduced LEC clones. This assay measures the uptake of a fluorescent substrate (MUH) into viable cells present in the well 72 h after addition of AM or vehicle control. Considering that the readout of the assay is 72 hours after onset of treatment, already minor differences in the number of LECs seeded in the wells can amplify and potentially lead to substantial differences in untreated LECs present in the wells 72 h later (as a result of cell proliferation). This is not a problem when pipetting LECs of the same origin (i.e. one type of shRNA-transduced LECs). However, we experienced that differences / variance may arise when seeding LECs from different shRNA-transduced LECs preparations. For this reason, we think that the procedure introduced a degree of inter-assay variability in the baseline values (cell numbers) between the shRNA-treated groups due to technical, and not necessarily biological reasons. Therefore, in our opinion, repeated, direct comparisons between all shRNA groups (and treatments) were not possible in this case. As this potential error appeared to average out with experimental repetitions, we nevertheless felt confident that we can perform a grouped analysis using repeated-measures ANOVA, by calculating the degree of proliferation induction in individual groups (after AM treatment). Hence, we calculated repeated pairwise comparisons between AM-treated and respective untreated control groups. When choosing this type of analysis, Tukey’s multiple comparisons test is not an option for post hoc analysis, as it requires the comparison of all mean values to every other mean value within one experiment. Instead, with this type of analysis (i.e. when comparing individual (mean) values to selected control (mean) values), Šídák’s multiple comparisons is one of the recommended posthoc tests. Thus, by choosing to make direct pairwise comparisons between AM-treated mean values with the respective untreated control mean values only, we circumvented that technical error/ variance could compromise the statistical analysis and were still able to show potential differences in % proliferation induction, relative to the untreated control. 

In conclusion, the reason why the results shown in Figure 4F are more significant than those in Figure S4C because a different statistical analysis was performed. We have now updated the Statistics paragraph in the Methods section, to better explain which test was performed for which type of data (lines 321-324 & 326-327). 

8) Statistical tests and p values should be provided in Supp Fig 1B and SF1G to support claims made in the manuscript.

Response: We have now performed a statistical analysis of the fold-change data reported in Figure S1B and S1G. Specifically, a One sample t test, comparing to the value of 1 was performed (Line 829). 

Minor: human gene names should be italicised all caps not lower case for annotation of genes in qPCR

Response: We thank the Reviewer for spotting this mistake in Supplementary Figures 1 & 5 and the corresponding Figure legends (lines 810 and 853 and following). We have now italicized all gene names. Regarding capitalization of letters, the rules in the literature and of PLOS are ambiguous. In most publications we checked, only the first letter of the gene name was capitalized. Therefore, we decided to adopt this writing style (e.g. Ackr3, Ramp2). 

Reviewer #2: In the study by Halin et Al., “Reassessing the adrenomedullin scavenging function of AKR3 in lymphatic endothelial cells” the role of ACKR3 as scavenger of the vascular peptide hormone adrenomedullin (AM) was clarified. In this work, in vitro experiments were performed to evaluate the uptake of AM in primary human Lymphatic Endothelial Cells (LECs) and in ACKR3-overexpressing Human Epithelial Kidney (HEK) cells. The aim of this work was to verify a competition between the normal ACKR3-ligand CXCL11/12 and AM, to evaluate if this receptor can influence/reduce the AM proliferative effects on LECs. The Authors conclude that the AM does not compete with CXCL11/12 for ACKR3 binding, but it binds their canonical receptor (CALCRL and PAMP2/3. In my opinion this finding may have a relevance in this field, but major revisions are required.

#2.1.) The background should be improved, experimental procedures should be better described and rearranged. For example, first cell culture (including HEK cells), second the treatments and silencing on LECs, then the plasmid construction and transfection, finally the assay.

Response: The order of the different sections of the Material & Methods has been revised according to the Reviewer’s suggestion. Moreover, we have further detailed certain parts, especially those concerning the plasmid generation and transfections (lines 192– 232) and the custom-synthesis of AM-AF568 (lines 246-258). 

A few questions:

#2.2.) Why three type of LECs were used?

Response: Three different human LEC sources were used to account for potential differences in ACKR3 expression and scavenging activity between LECs from different donors and/ or differences caused by donor age. We hypothesized that we might not see the same effect in LECs from a 58 year-old donor vs. a neonatal or a juvenile LEC donor (also considering the reported role of ACKR3/AM during development). A sentence explaining the rationale for the use of three different types of LECs has now been inserted in the introduction (lines 90-92). 

#2.3.) Why the coating was different between adLECs and nd- and jd-LECs?

Response: Depending on their isolation method and source (in our lab vs commercial supplier), different culturing protocols were used, and hence these primary human LECs were conditioned to different coatings from the start. Therefore, LEC types were kept with the type of coating they were used to / that was recommended for their culture in order to avoid causing any unexpected change for the cells during experiments. As a side note; in our lab we had performed chemokine uptakes with both types of coatings beforehand and observed no obvious differences.

#2.4.) The abbreviations should be uniformed and the same name should be used for the same protein, for example: ACKR3 was used throughout manuscript, but to indicate the antibody against this protein the CXCR/RDC-1 APC was used. The terms CXCL11/12-AF647 should be clarified as extended terms of CXCL; AF647 should be described as alexa flour 647; Atto 565 should be explained.

Response: We thank the Reviewer for spotting these inconsistencies. We had used the old names for ACKR3 (i.e. CXCR7 / RDC-1) for the antibodies in those cases where the antibodies had been exclusively described under these names in the literature / the company homepage. However, for ease of understanding, this has now been changed and these antibodies are called “anti-ACKR3” throughout the manuscript and their old names only listed in brackets in the Material & Methods section (lines 163 – 165). 

We have also now inserted an additional sentence describing the composition and function of the chimeric chemokine CXCL11/12 (lines 354-357). Moreover, the fluorophores attached to the different chemokines are now introduced in the Methods part in lines 260-262. 

#2.5.) In the introduction, CCX771 should be described as an antagonist of ACKR3.

Response: As requested by the Reviewer, the inhibitor CCX771 has now already been introduced in the Introduction (lines 92-94).

#2.6) The description of the plasmids in the section “construction of expression plasmids for HEK293 cell transfection” should match with the description in the section “HEK293 cell transfection…”

Response: The relevant sections have now been re-organized and expanded (lines 193 – 232). 

#2.7) In Figure1D, the error bars should be indicated in the graphic, which could be expanded to avoid the overlap between 37 °C ° and 37 °C + CCX771.

Response: Sparked by the comments of the Reviewer 1 and Reviewer 2, we have revised the format and order of the entire Figure 1. In the case of Figure 1D (former and current Figure 1D) we have now adapted it to the style of all other quantifications in manuscript and also in this Figure, namely bar plots with error bars and symbols for the individual data points. We have also solved the space problem by using colored bars and explaining te bars in a separate legend (same style as e.g. used in Figure 2). 

#2.7) Please check and harmonize all the figures. For example, in fig 2D CXCL11/12 is reported as CXCl11/12; cell types are indicated in fig 3A, but not in fig 2.

Response: Sparked by the Reviewer’s comment we have once again checked all Figure panels and have hopefully removed all remaining typos.

---

## [Editor Report · Decision Letter 1]

27 Apr 2023

Reassessing the adrenomedullin scavenging function of ACKR3 in lymphatic endothelial cells

PONE-D-23-00791R1

Dear Dr. Cornelia Halin,

We’re pleased to inform you that your manuscript has been judged scientifically suitable for publication and will be formally accepted for publication once it meets all outstanding technical requirements.

Kind regards,

Gabriella Lupo

Academic Editor

PLOS ONE
---

## [Editor Report · Acceptance letter]

19 May 2023

PONE-D-23-00791R1 

Reassessing the adrenomedullin scavenging function of ACKR3 in lymphatic endothelial cells 

Dear Dr. Halin:

I'm pleased to inform you that your manuscript has been deemed suitable for publication in PLOS ONE. Congratulations! Your manuscript is now with our production department. 

Kind regards, 

on behalf of

Prof Gabriella Lupo 

Academic Editor

PLOS ONE